

# Changes in bacterial viability after preparation and storage of fecal microbiota transplantation solution using equine feces

Julia A. Arantes[1], Rebecca Di Pietro[2], Mélanie Ratté[2], Luis G. Arroyo[3], Mathilde Leclère[4] and Marcio C. Costa[2]

[1] Department of Veterinary Medicine/Faculty of Animal Science and Food Engineering, Universidade de São Paulo, Pirassununga, São Paulo, Brazil
[2] Department of Veterinary Biomedical Sciences/Faculty of Veterinary Medicine, University of Montreal, Saint Hyacinthe, QC, Canada
[3] Department of Clinical Studies/Ontario Veterinary College, University of Guelph, Guelph, ON, Canada
[4] Department of Clinical Sciences/Faculty of Veterinary Medicine, University of Montreal, Saint-Hyacinthe, QC, Canada

Corresponding author
Marcio C. Costa,
marcio.costa@umontreal.ca

## ABSTRACT

**Background:** Fecal microbiota transplantation (FMT) has been used as a treatment option for horses (*Equus caballus*) with gastrointestinal diseases. Several preparation and conservation protocols to improve bacterial survival have been studied in other species.

**Methods:** This study aimed to evaluate the impact of oxygen exposure and different protectant solutions on bacterial viability before and after freezing using horse feces. Fecal samples from 10 healthy horses were aliquoted and diluted in cryoprotectant solutions containing antioxidants ($n = 40$) or 10% glycerol ($n = 40$). Half of the aliquots from each dilution condition were prepared inside an anaerobic chamber, while the other half were prepared under ambient air conditions. Each sample was also analyzed fresh and after freezing at $-20$ °C for 90 days. Bacterial viability was assessed using flow cytometry. A mixed linear model and the Friedman and Wilcoxon tests were used depending on data distribution.

**Results:** Freeze-thawing decreased bacterial viability by 47% (mean ± SD: 51 ± 27% before, 27 ± 8% after; $p < 0.001$). Glycerol was superior to the cryoprotectant after freezing (32 ± 8% glycerol, 24 ± 8% cryoprotectant; $p < 0.001$). Oxygen exposure did not affect viability ($p = 0.13$). There was no statistical difference between protectant solutions in fresh samples ($p = 0.16$).

**Conclusions:** Fresh FMT solutions may be better for treating horses with dysbiosis, but if freezing cannot be avoided, glycerol should be used to dilute feces.

# INTRODUCTION

The intestinal microbiota plays an essential role in the health maintenance of horses (*Equus caballus*) (*Costa & Weese, 2012*), and strict anaerobic species constitute a large portion of the equine intestinal microbiota (*Costa et al., 2012*; *Massacci et al., 2020*). To

improve procedures for prophylaxis, diagnostics, and therapeutics, it is crucial to understand better the equine microbiome and the factors affecting it (*Sanz, 2022*). Disruption of the bacterial balance (dysbiosis) in horses is present in several diseases, such as colitis and colic (*Costa et al., 2012*; *McKinney et al., 2021*; *Ayoub et al., 2022*; *Arnold et al., 2021*; *Weese et al., 2015*; *Salem et al., 2019*; *Stewart et al., 2019*).

Fecal microbiota transplantation (FMT) transposes the fecal microbiota from a healthy donor to a recipient. It has been used to correct dysbiosis in horses, but there is evidence that the procedure has limited capacity to change the microbiota of the distal intestinal tract of horses (*Boucher et al., 2024*). Therefore, efforts to increase the procedure's efficacy are necessary.

In human feces, freezing and oxygen exposure can decrease bacterial viability, demonstrating that using fresh feces and anaerobic processing could contribute to the retention of obligate anaerobes (*Papanicolas et al., 2019*; *Bellali et al., 2019*; *Shimizu et al., 2021*). Freezing and thawing can also reduce bacterial viability in equine feces (*Kopper et al., 2021*). Therefore, cryoprotectants such as glycerol are usually added to the FMT solution to prevent the formation of intracellular water crystals and bacterial death (*Koh, 2013*).

Cryoprotectants containing nutrients and antioxidants have been shown to increase bacterial survival during the freezing of human feces (*Cardona et al., 2012*; *Bellali et al., 2020*; *Costello et al., 2015*; *Bellali et al., 2019*). One study evaluating bacterial viability in the feces of a single horse reported no significant decrease in viability after mixing and oxygen exposure (*Loublier et al., 2023*).

Current recommendations in horses include freezing FMT solution mainly because it is impractical to maintain healthy donors free of medications and on a forage-based diet for this purpose (*Mullen et al., 2018*). In addition, frequent testing required to avoid transmission of infectious agents adds to the cost.

This study aimed to investigate bacterial viability after preparing FMT solutions for horses and compare different preservation protocols. In addition, it was hypothesized that a greater alpha diversity and specific genera of the donor's microbiota could be associated with greater viability.

## MATERIALS AND METHODS

This study was approved by the University of Montreal's Animal Use Ethics Committee (protocol number: 21-Rech-2106) and met all requirements for ethical animal care.

### Animals

Ten research horses (six mares and four geldings) belonging to the Faculty of Veterinary Medicine of the University of Montreal were used as donors. There were five Quarter Horses, two Paint Horses, one Thoroughbred, one Canadian and one crossbred with an average age of 14 years (range: 10 to 19) and average body weight of 538 kg (range: 483 to 622). Those were asthmatic horses that were kept in remission. None of them had gastrointestinal diseases and had not received antibiotics or other medications for at least

3 months before the study. All horses were housed together with access to grass pasture and fed haylage. They had *ad libitum* access to water and a salt block.

## Sample collection and experimental design

Fecal samples were collected directly from the rectum using clean rectal gloves and appropriate lubricant to reduce environmental contamination. They were then placed in air-sealed plastic bags and transferred within 10 min into an anaerobic chamber (BactronEZ; Sheldon Manufacturing Inc., Cornelius, OR, USA) for processing.

Samples were aliquoted following the design presented in Fig. 1.

The cryoprotectant was prepared according to *Bellali et al. (2020)*. It contained sucrose (10 g), sterile skimmed milk (10 g), trehalose (5 g), $CaCl_2$ (0.1 g), $MgCl_2$ (0.1 g), KOH (0.3/0.6 g), ascorbic acid (1 g), uric acid (0.4 g) and glutathione (0.1 g) diluted in 1 L of 10X phosphate-buffered saline (PBS) (*Bellali et al., 2020*). The glycerol solution contained 10% of glycerol and saline solution (NaCl 0.9%). Both solutions were autoclaved before the sample processing. The solutions (7.2 mL) were mixed with feces (1.8 g) to produce a fecal slurry of 25% (wt/vol).

A sample representing each environmental condition was promptly analyzed for bacterial viability within 10 min, and an aliquot was frozen at −20 °C to be analyzed after 90 days. The storage at −20 °C was used in this study to reproduce the conditions available at most Veterinary Hospitals.

## Bacterial viability processing and analyses

Bacterial viability of each condition (AnaeGly, AnaeCryo, AerGly and AerCryo) was determined using the LIVE/DEAD® BacLight™ Bacterial Viability and Counting Kit for flow cytometry (Waltham, MA, USA). Serial dilutions were made to choose the best for the optimal event rate for absolute enumeration. The commercially available bacterial viability and counting kit consists of two stains, propidium iodide (PI) and SYTO9, which both stain nucleic acids. Red fluorescing PI enters cells with damaged cytoplasmic membranes, whereas green fluorescing SYTO9 enters all cells and assesses total cell counts.

Untreated live cells and heat-treated cells from a fecal suspension were used to adjust the flow cytometer and optimize the dye ratio. A fecal sample diluted to a $10^{-3}$ dilution for optimal flow and event counts was stained with 1.5 µL of SYTO9 and 1.5 µL of PI, followed by a 15-min incubation in the dark at room temperature.

Data were acquired on a BD LSRFortessa™ X-20 Cell Analyzer using a fluorescence-activated cell sorter (FACS) Diva software v9.0 (BD Biosciences). The data were further analyzed using the FlowJo software v10.7.0 (FlowJo, LLC). The number of signals in the bacterial region (bac region) divided by the number of signals in the region of the beads provides the total number of bacteria per $10^{-6}$/mL in the flow cytometric analysis tube. Likewise, the number of events in the living bacteria area divided by the number of events in the cord region provides the number of live bacteria per $10^{-6}$/mL in the flow cytometric analysis tube. The concentration of both live and dead bacteria was determined using the following equation: ((# of events in bac. region) × (dilution factors))/((# of events in bead region) × $10^{-6}$) = bacteria/mL.

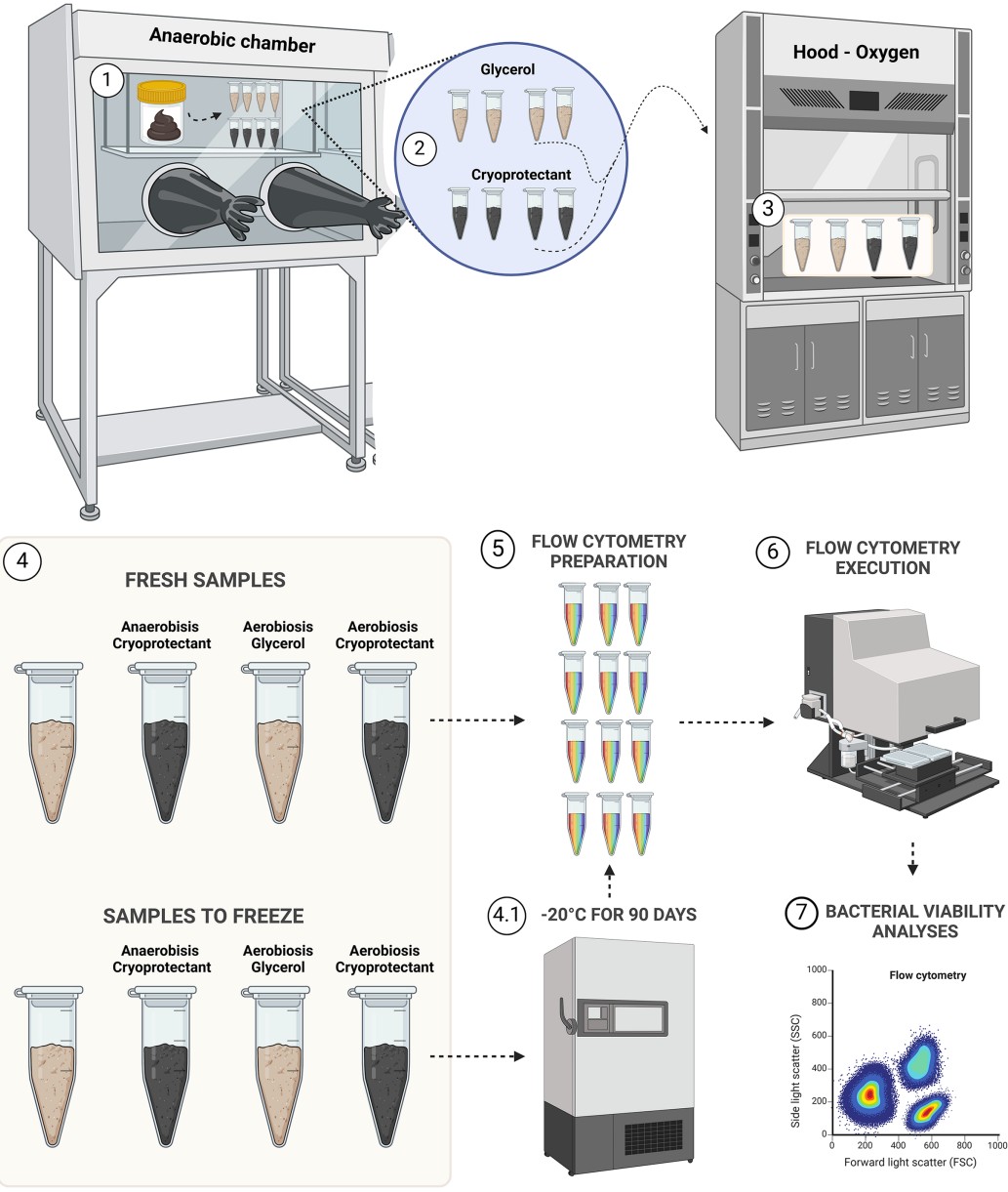

**Figure 1** **Experimental design for bacterial viability.** Step 1, transposition of feces to eight aliquots in microtubes; step 2, protectants mixing with fecal samples; step 3, aerobic exposition for aerobic groups; step 4, demonstration of the four groups of fresh samples (AnaeGly, AnaeCryo, AerGly and AerCryo) and the same for samples to freeze; step 4.1, illustration of the four groups that were frozen for 90 days; step 5, flow cytometry preparation in triplicate for all samples; step 6, flow cytometry execution; and step 7, flow cytometry bacterial viability analyses. Created with BioRender.com.

The staining procedures were performed in triplicate for each fecal sample under aerobic and anaerobic conditions. For anaerobic conditions, all staining procedures were performed inside the anaerobic chamber except for the acquisition process on the flow cytometer. The mean value of the triplicate was reported.

## Bacterial composition

The DNA of fresh fecal samples from all ten horses was extracted using the DNeasy PowerSoil Kit (Qiagen® Toronto, ON, Canada) following the manufacturer's instructions and stored at −20 °C. Amplification of the V4 region of the 16S rRNA gene was performed by polymerase chain reaction (PCR) at 95 °C for 3 min, followed by 35 cycles of 95 °C for 30 s, 55 °C for 90 s and 72 °C for 30 s, and final extension at 72 °C for 5 min. The oligonucleotide primers used were forward S-D-Bact-0564- a-S-15 (5′-AYTGGGYDTAAAGNG-3′) and reverse S-D-Bact- 0785-b-A-18 (5′-TACNVGGGTATCTAATCC-3′). Sequencing was performed using an Illumina MiSeq platform at the McGill University and Genome Quebec Innovation Centre.

Bioinformatics analysis was performed using the software Mothur (version 1.46.1) following the SOP recommended by *Kozich et al. (2013)*. Briefly, original fastq files were assembled into contigs, excluding sequences longer than 300 bp, those with base pair ambiguities, and those with homopolymers longer than 8 bp. The sequences were aligned using the SILVA 16S rRNA reference database (*Quast et al., 2013*). Chimeras were identified and removed, and sequences with 97% similarity were merged. The taxonomic classification was obtained from the RDP (Ribosomal Database Project) (*Cole et al., 2014*). Sequences classified as the same genus were grouped into phylotypes.

Microbiota analysis was performed to compare samples from horses with low (<60%) *vs*. high (>70%) bacterial viability. Alpha diversity was indicated by the number of genera, the Chao index (richness) and the Simpson and Shannon indices (diversity) (*Kozich et al., 2013*).

## Statistical analysis

Statistical analyses for bacterial viability were performed using R version 4.1.2 (*R Core Team, 2021*). A descriptive statistic was realized, and the bacterial viability was expressed as mean ± SD. The normality of data distribution was tested by the Shapiro-Wilk and the homoscedasticity by the Levene test. The Friedman test was used to analyze the effect for more than two groups when the data was non-normal. In cases of significant results, the Wilcoxon with correction of Bonferroni was performed. For normal data and more than two groups, a mixed linear model was used with viability as a dependent variable and atmospheric environment (aerobic x anaerobic) and protectants (glycerol x cryoprotectant) as fixed effects, with an interaction between the two variables to obtain the results for the four groups of frozen samples. Repeated measures structures (animals) were considered a random factor in these analyses. The Wilcoxon test was used to compare two groups with non-normal data.

The Pearson correlation test was used to investigate the correlation between viability and each condition (fresh and after freezing: AnaeGly, AnaeCryo, AerGly and AerCryo) and between viability and alfa diversity indices. In addition, the Molecular Analysis of Variance (AMOVA) was used to compare community composition and structure between samples with low and high viability.

To investigate the significance of relative abundances in samples with high and low viability, linear discriminant analysis effect size (LEfSe) was used (*Segata et al., 2011*). In

this case, the Kruskal-Wallis test was applied, a non-parametric test to detect differences between the groups, and then an unpaired Wilcoxon test, a classification sum test. A linear discriminant analysis (LDA) result greater than 2 was considered significant. A $p < 0.05$ was considered significant.

## RESULTS

The percentages of bacterial viability found in the different protocols are shown in Fig. 2. Overall, freezing significantly reduced the bacterial viability from 51 ± 27% to 27 ± 8% (Wilcoxon test: $p < 0.001$). There was no significant difference in viable bacteria among fresh samples (Wilcoxon with Bonferroni: $p = 0.16$) processed at different conditions and solutions (AnaerCryo; AnaerGly; AerCryo; AaerGly).

When comparing the different groups after freezing (Fig. 2C), samples diluted in glycerol exhibited the highest bacterial viability: 32 ± 8% compared to 24 ± 8% observed in samples diluted with the cryoprotectant (mixed linear model: $p < 0.001$). For the atmospheric condition, the viability percentage did not differ significantly (mixed linear model: $p = 0.14$). There was no interaction between the type of solution and oxygen exposure (mixed linear model: $p = 0.12$).

The results of bacterial viability obtained in each sample are presented in Table 1. There was no significant correlation between alpha diversity indices (number of genera, Chao, Simpson and Shannon) and bacterial viability (Pearson: all $p > 0.05$).

Among fresh samples, four animals had consistently high bacterial viability (between 71% and 86%), and six had low bacterial viability (between 35% and 58%). It is likely an individual microbiota characteristic of the horse since no evident methodological reason could be found. According to the LEfSe analysis, Mogibacterium was statistically associated with samples with low viability, and unclassified Erysipelotrichales and unclassified Burkholderiales with samples with high viability (LEfSe: $p < 0.05$; LDA score > 2). However, when comparing the difference in microbial composition between these two groups using the AMOVA test, i.e., the genera included in the samples of each group (membership) and the proportion of each of these genera (structure), there was no significant difference (AMOVA: $p = 0.12$ and $p = 0.26$, respectively).

Figure 3 presents the relative abundances at the genus level found in fresh feces from the ten healthy horses used in the study.

## DISCUSSION

The present study evaluated the impact of oxygen exposure and freezing on the bacterial viability in FMT solutions and compared two protocols for cryopreservation using horse feces. Freeze-thawing decreased bacterial viability by almost 50%, and solution of 10% glycerol could better preserve bacteria than a cryoprotectant. Oxygen exposure did not affect viability significantly.

Fecal microbiota transplantation has been used empirically to treat horses with diarrhea. Intragastric administration of the FMT solution is preferred in horses because it is unlikely that solutions administered via enema can travel the long route from the rectum to the large colon, especially in cases of diarrhea concomitant with increased peristalsis.

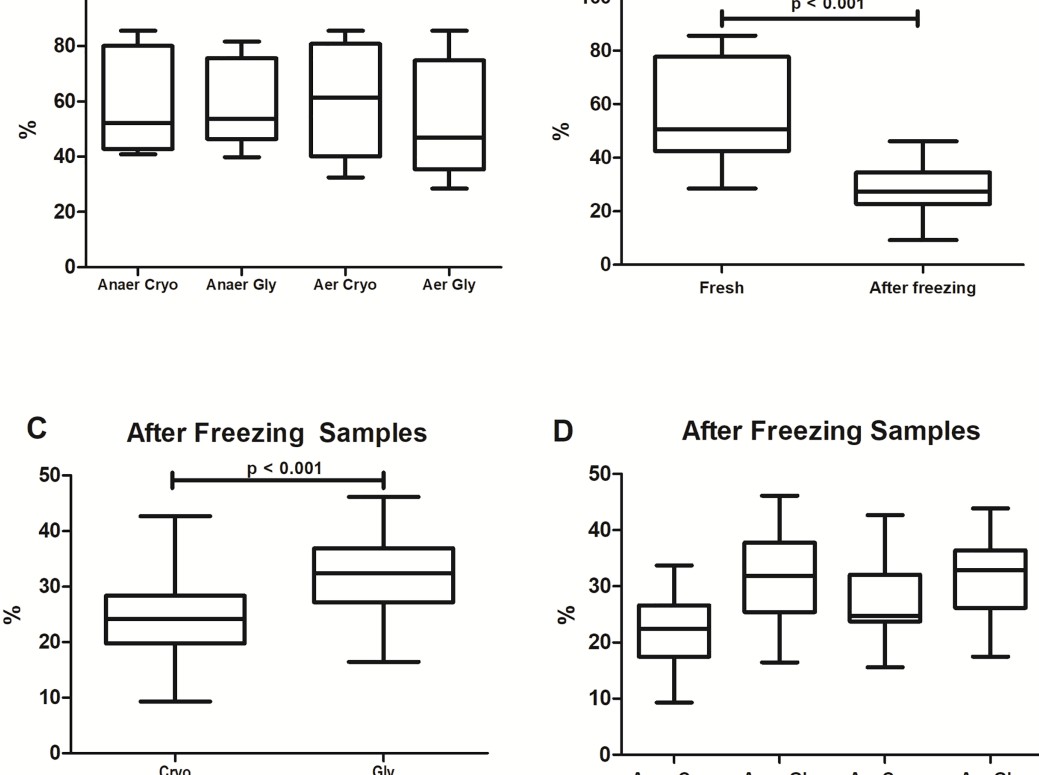

**Figure 2 Bacterial viability in equine fecal slurry, through the percentage of live bacteria, accessed by flow cytometry.** (A) Bacterial viability among fresh samples processed at different conditions: anaerobiosis + cryoprotectant (AnaeCryo); anaerobiosis + glycerol (AnaeGly); aerobiosis + cryoprotectant (AerCryo); aerobiosis + glycerol (AerGly). (B) Overall bacterial viability before (fresh) and after freezing. (C) Bacterial viability after freezing in samples diluted with a cryoprotectant (Cryo) or glycerol (Gly). (D) Bacterial viability after freezing of samples processed at different conditions (AnaeCryo, AnaeGly, AerCryo and AerGly). Note the different Y-axis ranges.               

However, exposure to gastric pH and digestive enzymes is associated with decreased bacterial viability in equine FMT solutions (*Kopper et al., 2021*). Therefore, improving the protocol of FMT preparation could better guide clinicians and ensure higher bacterial viability of the procedure.

The present study showed that freezing FMT fecal solutions for 90 days reduced bacterial viability by almost half compared to fresh samples. These results corroborate those of other studies using human feces, which also showed reduced viability after freezing (*Papanicolas et al., 2019*; *Bellali et al., 2020*). Those findings are especially important considering the current guidelines for horses recommend frozen FMT solutions (*Mullen et al., 2018*). Greater viability was reported when human feces samples were frozen at −70 °C compared to −20 °C (*Acha et al., 2005*; *Fowler & Toner, 2005*). Rapid freezing at lower temperatures seems to reduce bacterial damage, which may partially explain the marked decrease in viability in the present study. Since not all veterinary hospitals have ultra-low temperature freezers available for storage, −20 °C was used in this study to reproduce actual conditions applicable in a clinical setting.
**Table 1 Bacterial viability (in percentages) in fecal slurry of 10 horses before (Fresh) and after freezing assessed by flow cytometry.** Groups: anaerobiosis + cryoprotectant (AnaeCryo); anaerobiosis + glycerol (AnaeGly); aerobiosis + cryoprotectant (AerCryo); aerobiosis + glycerol (AerGly). Bold represents aliquots with high (>70%) viability.

| Horse | Fresh | | | | After freezing | | | |
|---|---|---|---|---|---|---|---|---|
| | AnaerGly (%) | AnaerCryo (%) | AerGly (%) | AerCryo (%) | AnaerGly (%) | AnaerCryo (%) | AerGly (%) | AerCryo (%) |
| 1 | 47 | 52 | 35 | **75** | 46 | 19 | 38 | 24 |
| 2 | 45 | 47 | 35 | 43 | 36 | 9 | 35 | 23 |
| 3 | **75** | **81** | **71** | **82** | 37 | 22 | 44 | 27 |
| 4 | **74** | **77** | **73** | 77 | 32 | 23 | 29 | 25 |
| 5 | **82** | **86** | **86** | **86** | 28 | 25 | 27 | 24 |
| 6 | **78** | **80** | **80** | **80** | 31 | 23 | 33 | 24 |
| 7 | 40 | 43 | 29 | 37 | 39 | 32 | 36 | 42 |
| 8 | 47 | 41 | 45 | 33 | 32 | 34 | 33 | 43 |
| 9 | 58 | 52 | 49 | 41 | 19 | 18 | 23 | 29 |
| 10 | 50 | 42 | 42 | 48 | 16 | 16 | 17 | 16 |
| Mean ± SD | 59.6 ± 15.1 | 60.1 ± 17.5 | 54.5 ± 19.9 | 60.2 ± 20.3 | 31.6 ± 8.5 | 22.1 ± 7 | 31.5 ± 7.4 | 27.7 ± 8.1 |

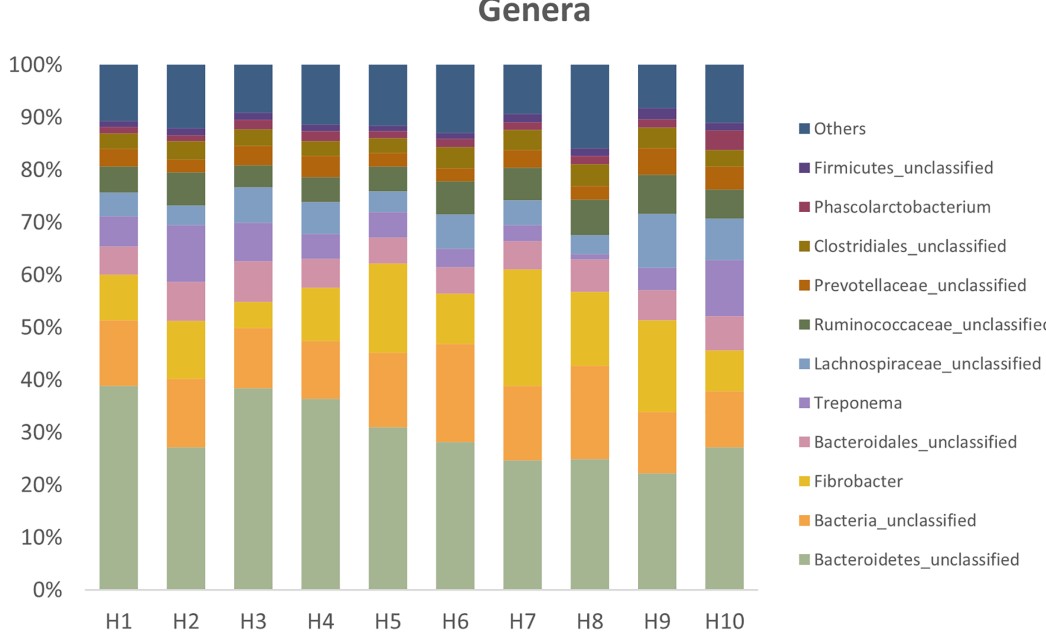

**Figure 3 Relative abundances at the genus level present in fresh feces from the ten healthy horses.** Horses 3, 4, 5 and 6 had high viability.

A cryoprotectant containing nutrients and antioxidants has been shown to preserve up to 84% of bacteria during freezing (*Bellali et al., 2020*). In the present study, no significant differences were found in the viability of fresh samples diluted with this cryoprotectant or glycerol. Still, glycerol was significantly more efficient in preventing bacterial damage after freezing. The reason why this protectant was not as efficient in horse feces remains to be

determined. Still, the different bacterial species constituting the microbiota of various animals may have different susceptibility to oxygen and damage caused by freezing (*Papanicolas et al., 2019*). For instance, the relatively slim lipid-rich cell wall of gram-negative bacteria may make them more vulnerable to the effects of freezing (*Kopper et al., 2021*). The amphiphilic nature of glycerol allows it to bond with hydrophilic and hydrophobic structures, offering broad cellular protection, which might have a wider coverage of the equine intestinal bacteria. Finally, the nutrients used for the cryoprotectant were designed to nourish components of the human microbiota, and the requirements of the horse's bacteria may differ.

Preparing FMT under anaerobic conditions has improved viability rates in human feces (*Papanicolas et al., 2019*; *Bellali et al., 2019*). Oxygen exposure would be expected to decrease bacterial viability (*Albenberg et al., 2014*), as strict anaerobic species constitute a large portion of the equine intestinal microbiota (*Costa et al., 2012*; *Massacci et al., 2020*). The present study evaluated the impact of oxygen exposure on the bacterial viability of horse feces for the first time. The comparison of FMT solutions prepared inside an anaerobic chamber with samples exposed to air revealed no significant differences. This can be considered a positive finding, as most facilities performing FMT in horses do not have easy access to those chambers. Furthermore, preparing large volumes inside a chamber can considerably increase the difficulty and processing time. Our findings agree with current guidelines for preparing FMT solutions using human (*Yadegar et al., 2024*; *Shi, 2020*) and dog (*Winston et al., 2024*) feces that preconized fecal manipulation at room air.

Although the results of the microbiota analysis showed high similarities in the main bacterial taxa of all donors, there was great variability in viability between individuals, varying from up to 86% to values as low as 40%. Similar viability variations have been reported among human donors (min 7.4; max 60.1%) (*Cibulkova et al., 2024*). This difference between donors suggests that differences in microbiota composition may result in the presence of bacteria that are more or less sensitive to FMT preparation. However, the search for bacterial taxa associated with greater viability revealed limited results, none among the most abundant bacteria in the donors. In fact, despite conventional DNA sequencing analysis providing taxonomic information, it does not inform whether the bacteria are dead or alive. Thus, the inclusion of bacterial viability in the selection of donors could be further explored to help identify horses that are potentially more efficient in restoring the intestinal ecosystem of patients with dysbiosis. Interestingly, both *Bell et al. (2024)* and *Long et al. (2024)* found a significant effect of the individual horse donor on microbiota composition after FMT processing, suggesting that the composition might also be necessary for donor selection. One limitation of the present study is the lack of microbiota analysis after FMT processing and storage.

*Di Pietro et al. (2023)* used horse feces to prepare regular and concentrated FMT solutions. In that study, feces from the donor horse were collected overnight and processed at room temperature before freezing. The microbiota analysis after sample manipulation showed significant compositional differences compared to fresh feces from the donors, including a marked increase of *Escherichia* spp. Similarly, *Martin De Bustamante et al.*

*(2021)* found significant compositional changes in the feces of 13 horses after 6 h at room temperature. The main changes included a decrease of Fibrobacteraceae and Ruminococcaceae after 6 h and an increase of Bacillaceae, Planococcaceae, Enterobacteriaceae and Moraxellaceae after 24 h. Conversely, *Bell et al. (2024)* used five horse donors to report no or minimum impact of FMT preparation and storing of FMT solutions at 4 °C for up to 72 h or at −20 °C for up to 28 days. Although those studies performed microbiota analysis, they did not investigate bacterial viability.

*Kopper et al. (2021)* used culture-based methods to assess bacterial viability following the current guidelines for FMT preparation. Freezing at −20 °C for 4 weeks and exposure to a simulated proximal gastrointestinal tract system significantly decreased bacterial viability, mainly affecting gram-negative bacteria (*Kopper et al., 2021*). In addition to the limitations of conventional culturing methods, which is the growth of only a small portion of the microbiota members, feces from a single horse were used in that study. *Loublier et al. (2023)* used the propidium monoazide (PMA) method, based on excluding non-viable DNA before 16S rRNA sequencing, to compare the microbial composition of FMT solutions made from manure of a single horse. They found no significant differences in the total living bacteria, but the relative abundances of some crucial components of the equine microbiota were affected by processing (*i.e.*, Fibrobacter, WCHB1-41_ge and Akkermansia).

*Long et al. (2024)* evaluated the impact of preparation and storage conditions of FMT solutions made from the feces of three equine donors. That study used a method comparing the sequencing of total DNA with complementary DNA (cDNA) to evaluate potentially metabolically active bacteria. That study reports that freezing significantly affected bacterial diversity of total DNA but not as much of cDNA, suggesting that potentially metabolically active bacteria are still present in the FMT solution. In agreement with the present study, they also found that glycerol was better than saline alone in preserving bacteria from freezing damage.

The present study reports a high percentage of non-viable bacteria consistently found among fresh samples (41% on average). Those organisms did not necessarily die between sampling and the analysis but constitute organic material from different portions of the intestinal tract, including many already dead bacteria. This information might be clinically relevant, as 40% of fecal bacteria might be already non-viable at the beginning of the FMT processing. Similar results have been reported in a human study (*Papanicolas et al., 2019*).

The methods employed in this study have certain limitations. Firstly, flow cytometry cannot effectively detect spores formed during exposure to oxygen without specific staining or the use of antibodies because of their small size (*Stopa, 2000*). This might be important considering that spore-forming species, such as Clostridia, constitute a significant fraction of the equine microbiota (*Costa et al., 2012*; *Massacci et al., 2020*; *Ericsson et al., 2016*). In addition, the study could have benefited from analyzing fresh samples diluted only in saline and not containing glycerol. However, there is no reason to believe that glycerol could affect bacterial viability in fresh samples, as it is used to avoid the formation of water crystals during the freezing process. Other studies evaluating the

dilution of horses and human feces in saline alone demonstrated similar results (*Long et al., 2024*; *Bellali et al., 2020*).

Together with the recent evidence from the literature, the results of the present study suggest that the administration of FMT to equine patients with dysbiosis is preferable to be given from fresh fecal slurries rather than frozen stored solutions. This raises concerns related to maintaining a healthy donor available at all times. That includes the frequency of testing against enteropathogens, avoiding variations in feed composition and factors that can change their microbiota (*e.g.*, antimicrobials, anthelmintics, stress, *etc.*). Therefore, if freezing is unavoidable, a 10% glycerol solution diluted in saline should be used, as the hypoosmolality of water could impact the bacterial viability of some species. Nevertheless, the clinical resolution of patients with recurrent *Clostridioides difficile* infection in humans is similar between fresh and frozen solutions (*Tang, Yin & Liu, 2017*) but might depend on the administered dose (*Agarwal et al., 2021*). Well-controlled clinical studies to address the use of FMT in horses are warranted (*Boucher et al., 2024*).

## CONCLUSIONS

Freezing markedly decreases bacterial viability in equine feces after preparing FMT solutions. If freezing is required, adding 10% glycerol could improve preservation. The interindividual variability among horses suggests that future longitudinal studies might reveal that viability could be included in selecting suitable donors. The current study adds new pieces of evidence to the controversial field of FMT administration in horses, suggesting that the current recommendations for preparing FMT solutions should be revised.

### Funding

This work was supported by the Equine Guelph (EG2022-01). The funders had no role in study design, data collection and analysis, decision to publish, or preparation of the manuscript.

### Grant Disclosures

The following grant information was disclosed by the authors:
Equine Guelph: EG2022-01.

### Competing Interests

The authors declare that they have no competing interests.

### Author Contributions

- Julia A. Arantes performed the experiments, analyzed the data, prepared figures and/or tables, authored or reviewed drafts of the article, and approved the final draft.
- Rebecca Di Pietro performed the experiments, prepared figures and/or tables, authored or reviewed drafts of the article, and approved the final draft.
- Mélanie Ratté performed the experiments, authored or reviewed drafts of the article, and approved the final draft.
- Luis G. Arroyo conceived and designed the experiments, authored or reviewed drafts of the article, and approved the final draft.
- Mathilde Leclère conceived and designed the experiments, authored or reviewed drafts of the article, and approved the final draft.
- Marcio C. Costa conceived and designed the experiments, analyzed the data, prepared figures and/or tables, authored or reviewed drafts of the article, and approved the final draft.

### Animal Ethics

The following information was supplied relating to ethical approvals (*i.e.*, approving body and any reference numbers):

University of Montreal's Animal Use Ethics Committee.

### DNA Deposition

The following information was supplied regarding the deposition of DNA sequences:

SRA accession numbers SRX26251449 to SRX26251458.

### Data Availability

The raw measurements of flow cytometry are available in the Supplemental Files.

### Supplemental Information

Supplemental information for this article can be found online at http://dx.doi.org/10.7717/peerj.18860#supplemental-information.

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
