# Peer review of "Changes in bacterial viability after preparation and storage of fecal microbiota transplantation solution using equine feces"

_PeerJ, doi:10.7717/peerj.18860_

## Round 0.1 · original submission · Major Revisions

A major revision decision has been made based on the reviewers' comments. Please respond to all the reviewers' comments in a specific and point-by-point manner. If you need more time to conduct extra experiments, please let us know so we can extend the revision time for you.

Reviewer 1 ·

Basic reporting

Clear and professional.

Experimental design
* * *
Validity of the findings
* * *
Additional comments

This study is researching methods for FMT to treat diseases in horses. This study is both very interesting and important.

This research shows that the process of anaerobic processes did not show significant meaning. In fact, in human clinical studies, there is a universally accepted method worldwide that does not recommend the use of anaerobic preparation. Please refer to the recent guidelines from China (PMID: 32701590) and US-Europe (PMID: 38717124), which do not involve anaerobic preparation. To enhance the credibility of the paper, it is suggested to discuss studies on human FMT, as there is sufficient evidence with an adequate sample size.

Reviewer 2 ·

Basic reporting

This is an interesting study with a clear rationale and some potential value in terms of future application in horse health. The work is well formatted, is grammatically sound, and contains professionally formatted figures. The work is well supported by a range of horse-specific and human-specific reference material and raw data are shared.

Experimental design

Experimental design is well explained and well considered. The work makes use of a robust experimental design - though just a few extra details may be needed in terms of repeatability for the selection of horses, horse diet and husbandry.
The ethics of the study are well considered and minimal interaction with living animals was required. There is no evidence of ethical issues.
Please provide the relevant test statistics alongside the p values in the work.

Validity of the findings

The findings from the work are clear - though there is scope to extend the explanation of the key findings in the discussion. As the work is relatively novel, it would be useful to provide some clear information on the next steps in this study area. Similarly, the conclusion could be expanded in terms of future application from a horse health perspective.

Additional comments

This is an interesting study and I commend the authors on their studies so far. I look forward to seeing a revised version of this manuscript.

Annotated reviews are not available for download in order to protect the identity of reviewers who chose to remain anonymous.

Reviewer 3 ·

Basic reporting

No comments

Experimental design

No comments

Validity of the findings

No comments

Additional comments

This manuscript aims to evaluate the impact of oxygen exposure and freezing on bacterial viability in fecal microbiota transplant (FMT) solutions for horses. However, several issues suggest the findings lack the necessary novelty and clarity to support publication. Numerious studies have already demonstrated the negative impact of oxygen exposure on anaerobic bacteria, particularly in equine and human FMT solutions, yet this study reports contradictory results without providing a clear explanation. Furthermore, the other findings presented—such as the superior effectiveness of glycerol as a cryoprotectant—have been extensively documented in prior studies, providing little new insight.

Without a robust analysis or explanation for the discrepancies in findings related to oxygen exposure, this study does not substantially advance the field. The conclusions offered are largely confirmatory and do not provide the clarity or novelty expected for publication. Consequently, I recommend rejecting this manuscript in its current form. The authors may consider revising their experimental design and conducting additional analyses to clarify these unexpected results.

---

## Round 0.2 · accepted · Accept

The reviewers' comments have been addressed by the authors, and the manuscript can be accepted for publication now.

Reviewer 2 ·

Basic reporting

This is an interesting study with a clear rationale and some potential value in terms of future application in horse health. The work is well formatted, is grammatically sound, and contains professionally formatted figures. The work is well supported by a range of horse-specific and human-specific reference material and raw data are shared.

Experimental design

Experimental design is well explained and well considered. The work makes use of a robust experimental design - though just a few extra details may be needed in terms of repeatability for the selection of horses, horse diet and husbandry.
The ethics of the study are well considered and minimal interaction with living animals was required. There is no evidence of ethical issues. In this revised copy, the required P values have now been included.

Validity of the findings

The findings from the work are clear - the authors have extended their explanations in the revised discussion. As the work is relatively novel, it would be useful to provide some clear information on the next steps in this study area. Similarly, the conclusion could be expanded in terms of future application from a horse health perspective.

Additional comments

Thank you for providing a clear and comprehensive revision of the manuscript, alongside a clear rebuttal letter. The work is now in a better position overall.